# Ferromagnetic soft catheter robots for minimally invasive bioprinting

Cheng Zhou[1,6], Youzhou Yang[1,6], Jiaxin Wang [2], Qingyang Wu[1], Zhuozhi Gu[1], Yuting Zhou[3], Xurui Liu[1], Yueying Yang[1], Hanchuan Tang [1], Qing Ling[2], Liu Wang [4✉] & Jianfeng Zang [1,5✉]

In vivo bioprinting has recently emerged as a direct fabrication technique to create artificial tissues and medical devices on target sites within the body, enabling advanced clinical strategies. However, existing in vivo bioprinting methods are often limited to applications near the skin or require open surgery for printing on internal organs. Here, we report a ferromagnetic soft catheter robot (FSCR) system capable of in situ computer-controlled bioprinting in a minimally invasive manner based on magnetic actuation. The FSCR is designed by dispersing ferromagnetic particles in a fiber-reinforced polymer matrix. This design results in stable ink extrusion and allows for printing various materials with different rheological properties and functionalities. A superimposed magnetic field drives the FSCR to achieve digitally controlled printing with high accuracy. We demonstrate printing multiple patterns on planar surfaces, and considering the non-planar surface of natural organs, we then develop an in situ printing strategy for curved surfaces and demonstrate minimally invasive in vivo bioprinting of hydrogels in a rat model. Our catheter robot will permit intelligent and minimally invasive bio-fabrication.

[1] School of Optical and Electronic Information and Wuhan National Laboratory for Optoelectronics, Huazhong University of Science and Technology, Wuhan, China. [2] Department of Urology, Tongji Hospital, Tongji Medical College, Huazhong University of Science and Technology, Wuhan, China. [3] The Key Laboratory of Bionic Engineering (Ministry of Education) and the College of Biological and Agricultural Engineering, Jilin University, Changchun, China. [4] CAS Key Laboratory of Mechanical Behavior and Design of Materials, Department of Modern Mechanics, University of Science and Technology of China, Hefei, Anhui, China. [5] The State Key Laboratory of Digital Manufacturing Equipment and Technology, Huazhong University of Science and Technology, Wuhan, China. [6] These authors contributed equally: Cheng Zhou, Youzhou Yang. ✉email: liuwang@mit.edu; jfzang@hust.edu.cn

The rapid development of three-dimensional (3D) printing has paved the way for myriad biomedical applications[1–5]. Driven by the development of implantable technology in healthcare[6,7], there is a growing interest in directly fabricating bio-tissue and/or biomedical devices on internal organs in living animals including humans. In vivo bioprinting that is capable of seamlessly integrating in situ printed materials and devices with the human body holds great promise in human tissue engineering and human-machine interface[8,9]. Currently, in vivo bioprinting is still in its infancy with most applications at or near the skin including, for example, skin or cartilage repair by direct ink-writing[10,11] or fabrication of epidermal electrodes[12]. For the printing on the internal organs of the human body, however, a surgical operation is usually required, which in turn poses a higher risk of infection and prolonged recovery time for patients. Therefore, minimally invasive bioprinting inside the body would be highly significant, but challenges remain. For example, attempts have been made to form patterned biopolymers by using a near-infrared light-induced polymerization under the skin. But the low penetrability of the light source limits the printing depth to around 5 mm[13–15]. Zhao et al. used a conventional motor-driven printer to directly write ink inside a chamber[16]. However, the nature of the rigid printer nozzle limits its application inside the body where tortuous anatomy is commonly encountered.

Recent advances in soft robots capable of dexterous manipulation have offered an opportunity to revolutionize surgical practice in a minimally invasive way[17–22]. For the minimally invasive operation that is characterized by a confined, easily deformable, dynamically changing environment, magnetoactive robots that can be remotely controlled to navigate hard-to-reach areas of the body have recently have garnered interest[23–27]. Due to the ease of untethered control, magnetic robots have broad potential applications including endovascular interventions and drug delivery to targeted lesions[28–31]. Among others, Kim et al. recently developed a ferromagnetic soft guidewire robot by uniformly dispersing ferromagnetic particles within a polymer matrix[32]. Upon magnetic actuation, such a robot can actively bend its tip and be swiftly steered through narrow and winding environments such as a brain vasculature phantom. In addition to omnidirectional steering and navigating capabilities, such a robot can be easily functionalized and integrated with other advanced technologies to permit more complicated biomedical applications.

Here, we report a ferromagnetic soft catheter robot (FSCR) system that is capable of minimally invasive in vivo bioprinting by incorporating magnetic actuation with 3D printing techniques. In the form of a slender rod-like structure with dispersed hard-magnetic particles, FSCR can reach regions inside the body using remote magnetic actuation, followed by in situ printing of functional inks (Fig. 1a) such as lesion healing creams and electrode gels. Distinct from conventional printing systems with a rigid nozzle (Fig. 1b), our FSCR feathers a magnetoactive soft nozzle that can print over a large workspace through a small incision (Supplementary Table 1). To realize steady extrusion of inks, our FSCR is rationally designed with an embedded reinforcing fiber mesh (Fig. 1c), which enables printing of various biocompatible and functional inks including silicones, silver pastes, and conductive hydrogels. A magnetic field is imposed by four numerically controlled motor-driven permanent magnets to achieve both translational and rotational motion of the FSCR (Fig. 1d). Compared with existing commercial apparatus for magnetically controllable catheters, the developed control system by employing four permanent magnets is relatively simple (Supplementary Table 2). Our FSCR can print different patterns using multiple inks on both flat and curved surfaces. We also demonstrate printing a functional hydrogel on a porcine tissue phantom and

the liver of a living rat in a minimally invasive, remotely controllable, and automated manner.

## Results
**Rational design of FSCR.** The FSCR is a slender rod-like structure with a hollow channel inside for material transport. The body of the FSCR is fabricated using an injection molding method, as illustrated in Fig. 2a. The ferromagnetic composite ink was first prepared by mixing uncured polymer resin (polydimethylsiloxane, PDMS) with evenly dispersed hard-magnetic microparticles (Neodymium iron boron, NdFeB). The composite ink was then injected into a tubular mold with a steel core wire placed at the center as the inner template. To enhance the mechanical performance of the FSCR, a polylactide (PLA) fiber mesh was inserted inside the mold. After being fully cured, the outer mold and inner wire were removed, producing a PLA fiber reinforced FSCR body with an inner channel for ink extrusion as highlighted by the green line in Fig. 2a. The details of the parameters for the PLA fiber reinforced FSCR can be found in Supplementary Fig. 1. To impart the desired magnetically responsive property of the robot, we magnetized the body by applying a strong impulse magnetic field (about 4 T) to magnetically saturate the dispersed NdFeB particles along its axial direction. When the applied magnetic field strength reaches 3 T, the residual magnetic flux density of FSCR tends to be saturated (Supplementary Fig. 2). Unlike soft magnetic materials that easily lose the induced magnetization once the external field is removed, hard-magnetic materials such as NdFeB, once magnetically saturated, retain their remanent magnetization due to their large coercivity. Therefore, the entire body of FSCR is characterized by a magnetization along its axial direction. The magnitude of the magnetization and Young's modulus of PDMS + NdFeB composite can be readily programmed by tuning the volume fraction of NdFeB inside the composite[32]. We employed a 15 vol.% of NdFeB, which confers a magnetization of 100 kA/m after full magnetization with Young's modulus of 1.15 MPa. By employing different molds, we can easily fabricate FSCRs of various sizes that can be used in different applications (Fig. 2a) and the smallest outer and inner diameter of our FSCR can be achieved as 2 and 0.6 mm, respectively, complying with the standard of incision size in minimally invasive surgeries[33]. In addition, the cured NdFeB + PDMS composite has no toxicity based on a cell viability test in which the cell survival rate is 98.6%, suggesting high biocompatibility of our FSCR according to the standard (70% cell survival rate) of USP (ISO 10993-5) (Supplementary Fig. 3)[32,34].

Our FSCR features an embedded PLA reinforcing mesh primarily to enhance the printing performance by restraining the lateral expansion of the printing channel when pressurized (Fig. 2b and Supplementary Movie 1). In general, due to the ink viscosity and friction between the ink and the nozzle, the input energy dissipates during the process of ink extrusion, leading to a significant pressure loss along the channel (see Supplementary Materials for details)[35]. Thus, a steady extrusion of the printing ink requires an applied pressure in the range of hundreds of kilopascals which would result in a substantial expansion in the diameter of the printing channel without enforcement, yielding a delay time after pressure is applied till the materials is extruded. To visualize lateral expansion, we prepared printing nozzles using pure PDMS with the same Young's modulus ($E = 1.15$ MPa, fabricated by varying the curing temperature and base-to-curing agent weight ratio) as that of PDMS + NdFeB composite and dyed the printing ink with orange coloring (Fig. 2b, see "Methods" section for details). The performance of both reinforced and non-reinforced designs under various applied pressures was compared. Figure 2c clearly suggests that the lateral expansion, characterized by $D/d$ where $d$ and $D$ are the channel

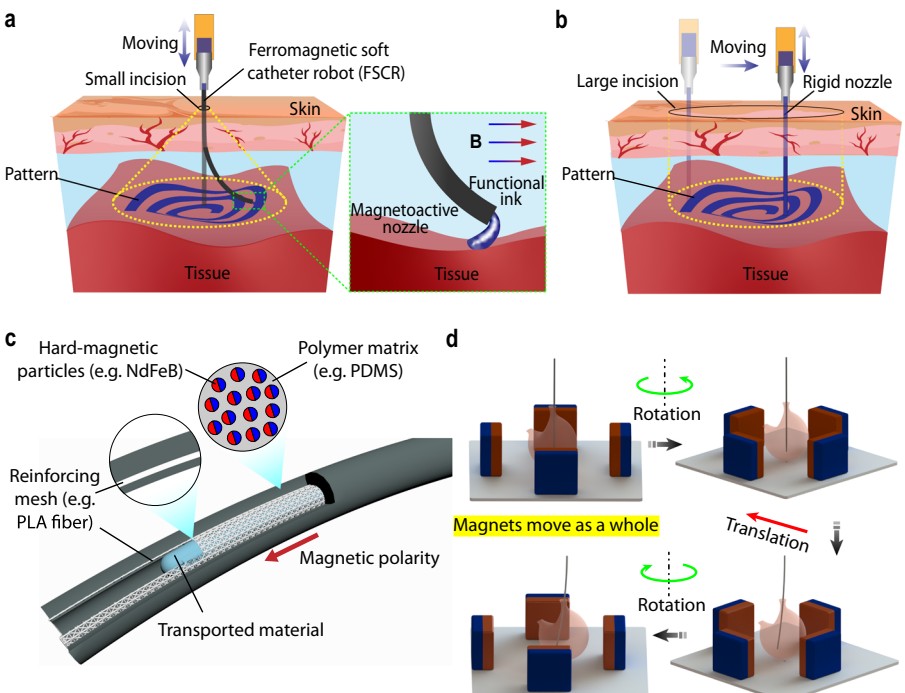

**Fig. 1 Ferromagnetic soft catheter robot (FSCR) system for minimally invasive bioprinting. a** Schematic illustration of the minimally invasive printing with functional inks (e.g., conducting polymer and living materials) inside the human body through a small incision; in the zoomed-in panel, **B** represents the magnetic field. **b** Schematic illustration of the traditional printing system using a rigid nozzle with a large incision. **c** Schematic of FSCR that is composed of soft polymer matrix with dispersed hard-magnetic particles and polylactide (PLA) reinforcing mesh. The magnetic polarity is programmed along the FSCR axial direction. **d** The numerical control strategy of FSCR, where operations are instructed by digital data. Maneuvering of the FSCR via rotation and translation of the four permanent magnets.

diameter before and after ink extrusion, respectively, of the non-reinforced sample is much higher than the reinforced counterpart. As a result, non-reinforced samples not only show an impaired printing resolution but also have a slower flow rate (Supplementary Fig. 4), giving rise to a markedly increased delay time during printing (Fig. 2d)[36]. By contrast, the reinforced design exhibits a small lateral expansion of 4% when the pressure is increased to 240 kPa and also maintains steady extrusion over time (Fig. 2e). Since delay time is longer when FSCRs are made of softer materials (Supplementary Fig. 5), the PLA reinforcing mesh is required. And the mechanical testing of fabricated reinforced and non-reinforced FSCR is shown in Supplementary Fig. 6.

It is also worth noting that the presence of the reinforcing mesh has a minimal influence on the bending behavior of the FSCR when it is subjected to magnetic fields. In this regard, we compared the tip deflection $\delta$ of the FSCR when approaching a cuboidal magnet ($50 \times 50 \times 30$ mm with surface induction of about 400 mT). The experimental setup is depicted in Fig. 2f, in which the robot tip initially falls in the central line of the magnet. Note that such a cuboidal magnet is later used in building our magnetic control system. We found that the reinforced FSCR can easily bend up to $\delta/L = 0.3$ where $L$ denotes the robot's length (Fig. 2f); this provides a large enough workspace for magnetically controlled printing. Only small differences have been observed in the bending performance between reinforced and non-reinforced cases upon magnetic actuation. Therefore, the embedded PLA reinforcing mesh provides the FSCR with steady printing performance upon magnetic actuation.

**Magnetically-controlled printing system.** To realize an automated printing process, we customized a magnetically controlled printing system utilizing a set of motors that can be numerically controlled by a computer. Figure 3a illustrates the printing system

apparatus that consists of a fixed printing platform, an FSCR printing nozzle, and four cuboidal magnets. The normal vector of the printing platform at its center defines the $z$-axis along which the printing nozzle can be moved up and down by a motor. Four magnets are placed in a rectangular layout with the north pole facing inside (Supplementary Fig. 7) and their symmetric planes (i.e., central planes) define the XY plane and XZ plane of the coordinate system, as highlighted by the yellow and green box. The motors drive the four magnets to move in a concerted fashion providing both translational displacement along $x$-direction and rotation about the $z$-axis, denoted as $T_{mag}$ and $\theta_{mag}$, respectively. To achieve steady printing, the nozzle tip is always placed at XY plane at a fixed distance from the printing plane that is equal to FSCR's printing linewidth ($0.6\sim1$ mm).

The superimposed magnetic field generated by four cuboidal magnets is nonuniform. The distribution of magnetic flux density $\mathbf{B}=(B_x, B_y, B_z)$ for such a nonuniform field was measured using a 3D Hall probe, as presented in Fig. 3b and Supplementary Fig. 8a. In a nonuniform magnetic field, hard-magnetic materials experience both magnetic torque and body force:

$$\boldsymbol{\tau} = \mathbf{M} \times \mathbf{B} \tag{1}$$

$$\boldsymbol{f} = (\mathbf{M} \cdot \nabla)\mathbf{B} \tag{2}$$

where $\mathbf{M}$ is the magnetic moment density (magnetization) of the material, $\mathbf{B}$ is the total magnetic field from the magnet and $\nabla \mathbf{B}$ describes the magnetic field gradient. The FSCR has a magnetic moment along its axial direction, i.e., negative z-direction in Fig. 3a. The primary reasons for designing such a 4-magnet control system are twofold. First, due to symmetry, the magnetic field has $B_x = B_y = 0$ along the z-axis and in particular $\mathbf{B} = \mathbf{0}$ at the origin. Thus, the FSCR is at equilibrium initially when the entire body is aligned with the z-axis and the tip coincides with

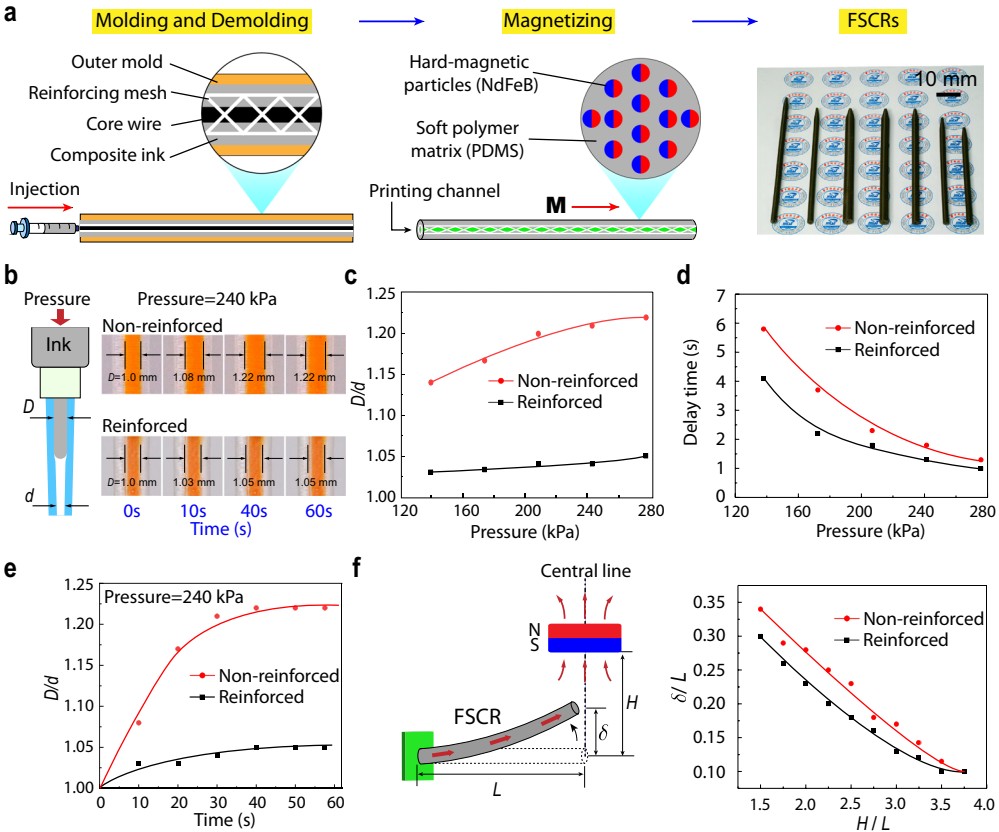

**Fig. 2 Design of ferromagnetic soft catheter robot. a** The fabrication process of PLA-reinforced FSCR using an injection molding method. The magnetic polarity of FSCRs denoted as **M**, is along its axial direction after being magnetized to saturation by a strong impulse magnetic field. FSCRs with various diameters (4–7 mm) and lengths (80–110 mm) are shown on the right. **b** Sketch of printing ink (colored orange) extrusion process in an FSCR under applied pressure (left). $L$ is the robot length. $d$ and $D$ are the diameter of the printing channel before and after ink extrusion, respectively. Experimental pictures of both reinforced and non-reinforced catheters ($L = 100$ mm, $d = 1$ mm, catheter outer diameter 4 mm) under the pressure of 240 kPa (right). **c** Diameter expansion ratio $D/d$ under various applied pressures for the reinforced and non-reinforced catheters. **d** Delay time as a function of applied pressure for the reinforced and non-reinforced catheters. **e** Plots of diameter expansion ratio $D/d$ measured over times under the pressure of 240 kPa. **f** Schematic of FSCR bending under different magnetic fields generated by a single cubic permanent magnet (left). Experimental measurements of normalized deflection $\delta/L$ of non-reinforced and reinforced catheter plotted against the normalized actuating distance $H/L$ (right).

the origin, maintaining a vertical configuration before printing. Second, placing two magnets with north poles facing each other in both $x$-direction and $y$-direction allow for more stable control of the FSCR. The translational and rotational displacement of 4 magnets as a whole will alter the magnetic field in space and thus drive the FSCR to bend and rotate, giving rise to a translational displacement, $T_{tip}$, and a rotational displacement, $\theta_{tip}$, of its tip, respectively (Fig. 3c). The maximum value of $T_{tip}$ determines the effective printing workspace, which varies with the length-to-diameter aspect ratio of FSCR (Supplementary Fig. 9). It is worth noting that the tip will also undergo an upward displacement from the XY plane, denoted as $U_{tip}$, as it translates outward (Fig. 3d). Therefore, to compensate for the deviation from the XY plane, the FSCR should undergo a downward displacement by the motor with a magnitude identical to $U_{tip}$ (Fig. 3c, d). Since the relationship between magnet displacement and tip displacement is the foundation of digitally controlled printing, we first investigate this relationship using both finite element modeling (FEM) and experiments.

The body of the FSCR is made by uniformly dispersing micron-sized ferromagnetic particles (~5 μm) in the soft polymer, referred to as hard-magnetic soft materials[37,38]. Applying a nonuniform magnetic field induces magnetic torques and forces on the embedded ferromagnetic particles, which produces microscopic stresses that drive the macroscale deformation. Such microscopic stresses, denoted as magnetic Cauchy stresses, can be expressed as:

$$\boldsymbol{\sigma} = -\mathbf{B} \otimes \mathbf{FM} \qquad (3)$$

where $\mathbf{F}$ is the deformation gradient and the $\otimes$ operation denotes the dyadic product which takes two vectors to yield a second-order tensor. Implementing the magnetic Cauchy stress in a user-defined element subroutine in the commercial finite-element software ABAQUS, we can simulate the deformation of FSCR within the nonuniform magnetic field. Note that the analytical form of such a nonuniform magnetic field was used in FEM that was derived from[39] and validated by our experimental measurements (Supplementary Fig. 8b). Simulated results for one FSCR with a length-to-diameter aspect ratio of 25 are presented in Fig. 3f–h and are in excellent agreement with the experimental data. Correlations between magnet displacements and tip displacements are found as $T_{mag} = 0.63T_{tip}$, $U_{tip} = 0.0028T_{mag}^2 - 0.007T_{mag}$, and $\theta_{tip} = \theta_{mag}$ (Fig. 3e). Note that when the printing platform is planar, the motor-driven FSCR will be pushed downward by a displacement equal to $U_{tip}$ to compensate for the deviation from the printing plane. For the printing of multilayered structures or on 3D non-planar surfaces, such a compensational displacement should be altered accordingly by taking into account the 3D surface morphology, i.e., the altitude variation in the z-direction. By mapping these relationships into the computational code, we can precisely manipulate the tip motion by

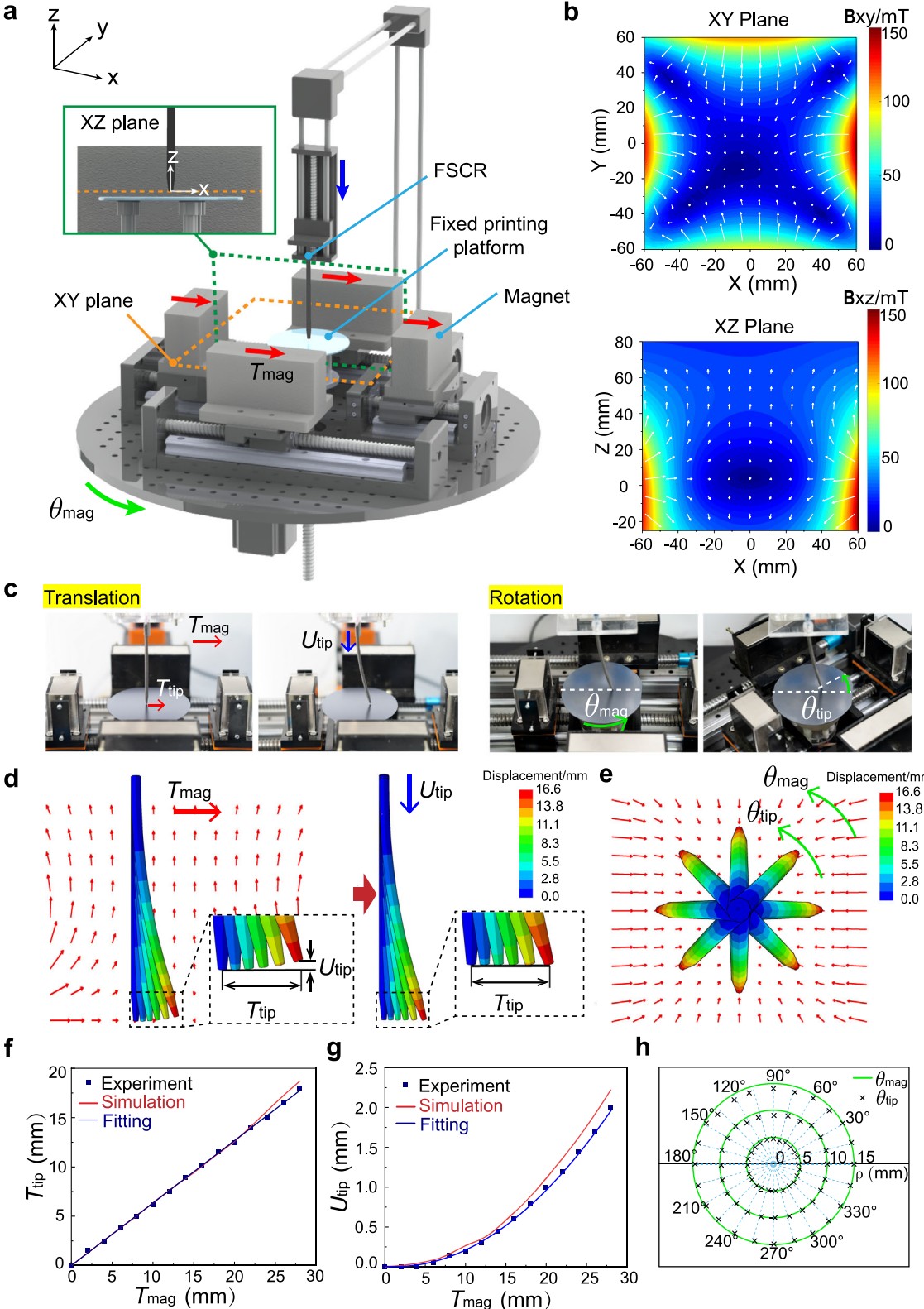

digitally controlling the movement of magnets, thus accomplishing magnetically-driven printing of various complex structures.

**Minimally invasive printing.** The magnetically controlled printing system can print various patterns on both a planar surface and non-planar surfaces, or even through a minimally invasive manner. To print, the target pattern needs to be converted into catheter-path codes according to the established relationship between $T_{tip}$, $U_{tip}$, and $T_{mag}$ (see "Methods" section for printing process details). The FSCR printing system is able to print PDMS-1700 and Ecoflex composite ink (viscosity ~340 Pa·s, Supplementary Fig. 10) into various patterns as demonstrated by a flower with six petals, a square spiral, a 3D tube, and a 3D scaffold (Fig. 4a, b, and Supplementary Movies 2–5)[39]. The printing process can be completed in a single stroke (e.g., the

**Fig. 3 Numerical control of FSCR for printing. a** Schematic illustration of the magnetically-controlled printing system. **b** Contour plots of magnetic flux density **B** in XY plane (up) and XZ plane (down) of the superimposed magnetic field, corresponding to the yellow and green boxes in **a**. The arrows indicate the magnetic field vectors and the background color represents the magnetic field strength as indicated by the color bar in mT. **c** Images showing the motion control of FSCR by moving four permanent magnets with both translational displacement along the x-axis and rotational displacement about the z-axis. For translational mode: as the magnets translate in the x-axis direction (denoted by $T_{mag}$), the tip of the FSCR moves to the same direction (denoted by $T_{tip}$); A downward displacement (denoted by $U_{tip}$) compensates the lift of the XY plane during the translation. For rotational mode: the magnetic field is rotated by $\theta_{mag}$; Tip of the FSCR is rotated to the same direction by $\theta_{tip}$. **d** Simulation of the translation process. Left panel: computational $T_{tip}$ when magnetic field translated by $T_{mag}$. Right panel: computational compensation of $U_{tip}$ as $T_{tip}$ increases. The color represents the displacement magnitude. **e** Simulation of the rotation process. The eight states show that when the magnets rotate, the tip of the FSCR rotates at the same angle. The color represents the displacement magnitude. **f** Experimental and simulated results for the horizontal displacement of the FSCR tip as a function of magnet displacement $T_{tip} = 0.63T_{mag}$. **g** Experimental and simulated results for the vertical displacement of the tip as a function of magnet displacement $U_{tip} = 0.0028T_{mag}^2$-$0.007T_{mag}$. **h** Experimental results for the rotational displacement of the FSCR tip as a function of magnet rotation, $\theta_{tip} = \theta_{mag}$.

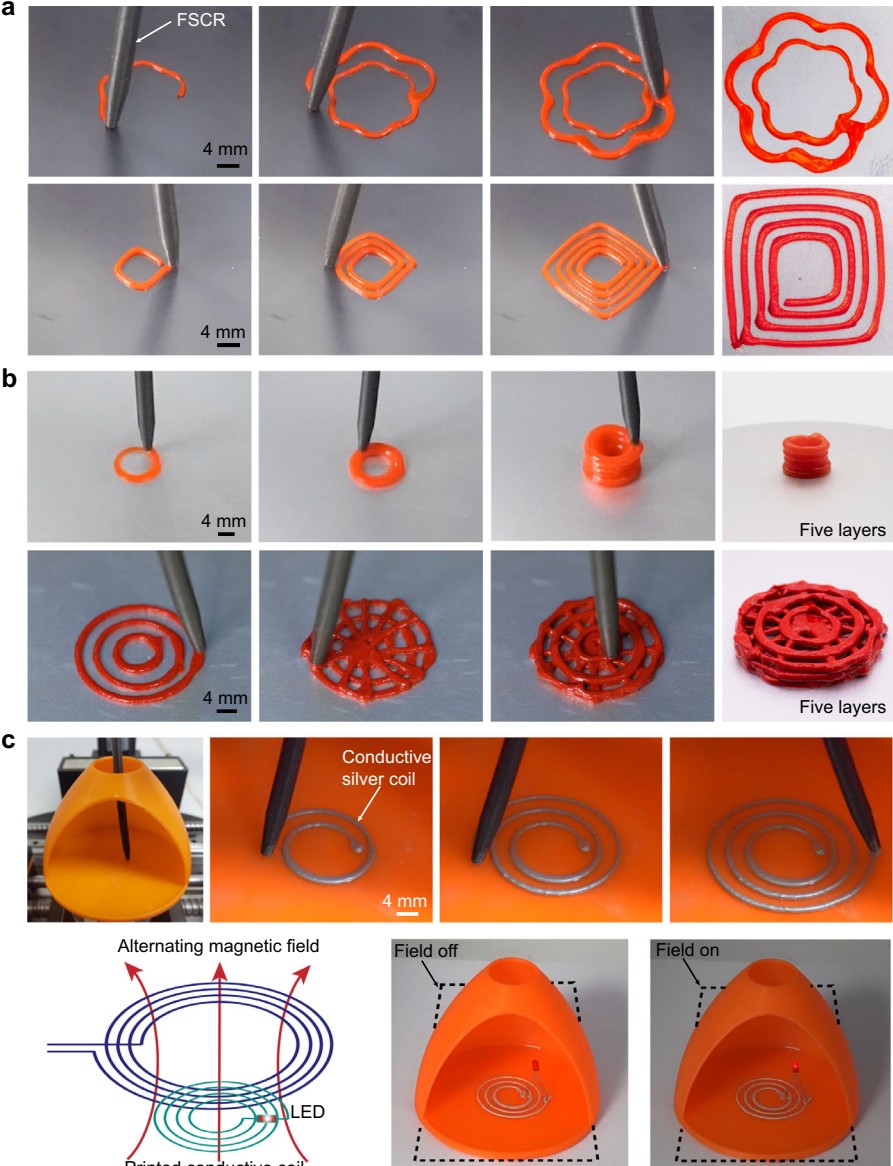

**Fig. 4 Direct ink printing by an FSCR. a** Demonstration of FSCR printing with viscoelastic materials (PDMS-1700 and Ecoflex composite) into different patterns such as a flower with six petals and a square spiral. **b** Printing 3D five-layered tube and scaffold with viscoelastic materials. **c** Upper panels: Images of experimental demonstration of printing functional device. An LED was placed as an energy-dissipating element on a conductive silver ink coil after printing. Bottom panels: left, schematic of the printed wireless device that powered a surface-mounted LED under an alternating magnetic field from an electromagnetic coil (power: 1600 watts) at a distance of 10 mm. Middle: the actual circuit. Right: LED activation upon imposing and alternating magnetic field.

flower, square spiral pattern, 3D tube) or multiple strokes with different initial positions of the printing nozzle (e.g., the 3D scaffold). As presented in Supplementary Fig. 11, all printed patterns exhibit excellent agreement with the original designs, demonstrating a high printing accuracy of the FSCR. Due to the viscoelastic nature, the extruded ink usually has a die-swelling phenomenon (Supplementary Fig. 12a), resulting in a printed fiber with a diameter $\alpha d$, where $d$ is the inner diameter of the printing nozzle and $\alpha$ is the swelling ratio[40,41]. The resolution of the printed fiber (i.e., $\alpha d$) of our FSCR mainly depends on four parameters: the moving velocity of the nozzle, the input pressure, the inner diameter of the nozzle $d$, and the viscosity of the ink. As shown in Supplementary Fig. 12b, a faster-moving velocity usually stretches the printed fiber, leading to a smaller $\alpha d$; while increasing the input pressure, and the size of the inner diameter and the viscosity will increase $\alpha d$. Note that to ensure a continuous printed fiber without accumulation or discontinuity, the velocity of the nozzle should be well controlled[40,41]. Based on the injection molding method, the smallest inner diameter $d$ was achieved as 0.6 mm, which yields a resolution $\alpha d$ of 0.53 mm at the moving velocity of 3.3 mm/s, the pressure of 240 kPa, and the viscosity of 339 Pa·s (Supplementary Fig. 12b, c).

Given the soft and slender nature of the FSCR, it can be threaded through a small aperture and programmed to print a wireless electronic device into a spiral pattern on the bottom of a chamber with conductive ink, as shown in Fig. 4c. The conductive ink is composed of silver flakes in an alginate solution with an added trace ethanol[12]. The resistance of the composite conductive ink can be changed by varying the weight fraction of silver flakes from 68.7% to 93% in dry conditions (Supplementary Fig. 13). The spiral conductive coil can be connected with an electronic component such as a commercial light-emitting diode (LED). When actuated by an alternating magnetic field from an electromagnetic coil, the printed spiral wire can light up the LED wirelessly through electromagnetic induction (Fig. 4c and Supplementary Movie 6). In addition to the demonstrated printing ability, our FSCR is also capable of object manipulation in a minimally invasive manner. For example, it can deliver, move or suck out targeted materials either in liquid or in solid form with different shapes and variable weight (0.5–5 g) in confined environments (Supplementary Fig. 14a, b). Such an object manipulation capability allows for more applications of our FSCR in the future minimally invasive operations (Supplementary Movie 7 and 8). To demonstrate the capability of drug delivery to the target lesion, we carried out an experiment in which a hydrogel containing acetylsalicylic acid (ASA) was printed to a porcine tissue and validated the released drug by UV-vis spectrophotometer (see "Methods" for details, Supplementary Fig. 14c, d)[42].

**In vitro minimally invasive bioprinting**. To demonstrate the potential application of the FSCR in minimally invasive bioprinting, we printed a spiral pattern on an excised porcine tissue with a naturally non-planar surface using a minimal incision in an artificial skin overlaying the porcine tissue (Fig. 5a). To print the desired pattern, we need to identify a 3D path on the curved surface of the tissue to guide the nozzle tip. Using a digital scanner (Fig. 5b), the tissue surface was first reconstructed into a 3D model with $(x, y, z)$ coordinates data set (Supplementary Fig. 15) from which the printing path for the desired pattern was designed. By mapping such a printing path with the control parameters (Fig. 5c), $T_{mag}$ and $\theta_{mag}$, we generated the code to guide the printing nozzle. The FSCR was inserted through a small incision (diameter ~ 0.8 cm) in the artificial skin and a spiral pattern of a conductive hydrogel was printed at the surface of the

porcine tissue along the pre-defined path (Fig. 5d and Supplementary Movie 9). The entire process was completed within 2 min. Besides, the printed conductive hydrogel was also characterized by electrochemical impedance properties (Supplementary Fig. 16) and adhesion performance on the tissue surface (Supplementary Fig. 17).

**In vivo minimally invasive bioprinting**. We then evaluated the feasibility of minimally invasive bioprinting on a rat liver in vivo. First, computed tomography (CT) technology was utilized to reconstruct the 3D surface of the liver in a living rat (Fig. 6a). The reconstructed 3D model and the upper surface of the liver after extraction and smoothing are shown in Supplementary Fig. 18. A printing path was then defined on the upper surface of the liver, and the control code was generated to guide the printing nozzle (Fig. 6b). To clearly demonstrate the printing process, the rat abdomen was continuously insufflated with carbon dioxide to provide a large and stable operating space, and a digital laparoscope (diameter 0.5 cm) was inserted to record the printing process through a minimal cut in the abdomen (Fig. 6c). In this demonstration, a thinner FSCR (25 mm in length and 2 mm in diameter) was employed because of the confined space in the rat abdominal cavity. As presented in Fig. 6d, an Archimedes spiral pattern (material: conductive hydrogel) was successfully printed at the surface of the liver through a small incision (diameter ~3 mm) within 70 s (Supplementary Movie 10).

## Discussion

In this work, we introduced a minimally invasive ferromagnetic soft catheter robot and developed a printing system that can be remotely controlled by a computer. We have demonstrated our proof-of-concept studies by printing various patterns using different functional inks on both flat and naturally curved surfaces and succeeded in all cases. Overall, the soft catheter robot has distinct advantages when working in a confined space in a minimally invasive manner compared with conventional robots (as shown in Supplementary Table 2). For future applications[43–48], we propose a digital control strategy utilizing magnetic actuation that would allow surgeons to complete operations away from x-ray radiation.

This minimally invasive in vivo bioprinting technology is still in its infancy there will be limitations regarding printing speed, resolution, and complexity of the printed pattern. To adapt to both complex three-dimensional patterns and the confined biological environment, further optimization of the magnetic domain and the miniaturization of the body of the FSCR body will be needed[49]. In addition, a more versatile magnetic field can be designed; thus, for instance, the four permanent magnet setups can be upgraded to a 6-polar-magnet system, which allows for more freedom of control[50]. The current printing system adopts the CT to reconstruct the 3D topology of the tissue that is later used to generate the numerical code to print. In the future, utilizing the intraoperative CT and/or equipping the robot with vision-based sensors (e.g., stereovision and structured-light scanning[51]), the real-time tomography of the tissue can be constructed, and augmented reality for real-time bioprinting can be achieved[52]. In this regard, a close-loop soft robotic system with feedback based on real-time imaging may further improve the accuracy of printing[53]. The coding system to guide the printing path can also be optimized to create even more complicated patterns and 3D architectures with high resolution. Moreover, progress on functional materials (e.g., biomaterials for gastric ulcer healing[54] and/or health monitoring[55]) with bioprinting-compatible properties (e.g., rheology and adhesion) will enable the FSCR to print more complicated 3D patterns/architectures to

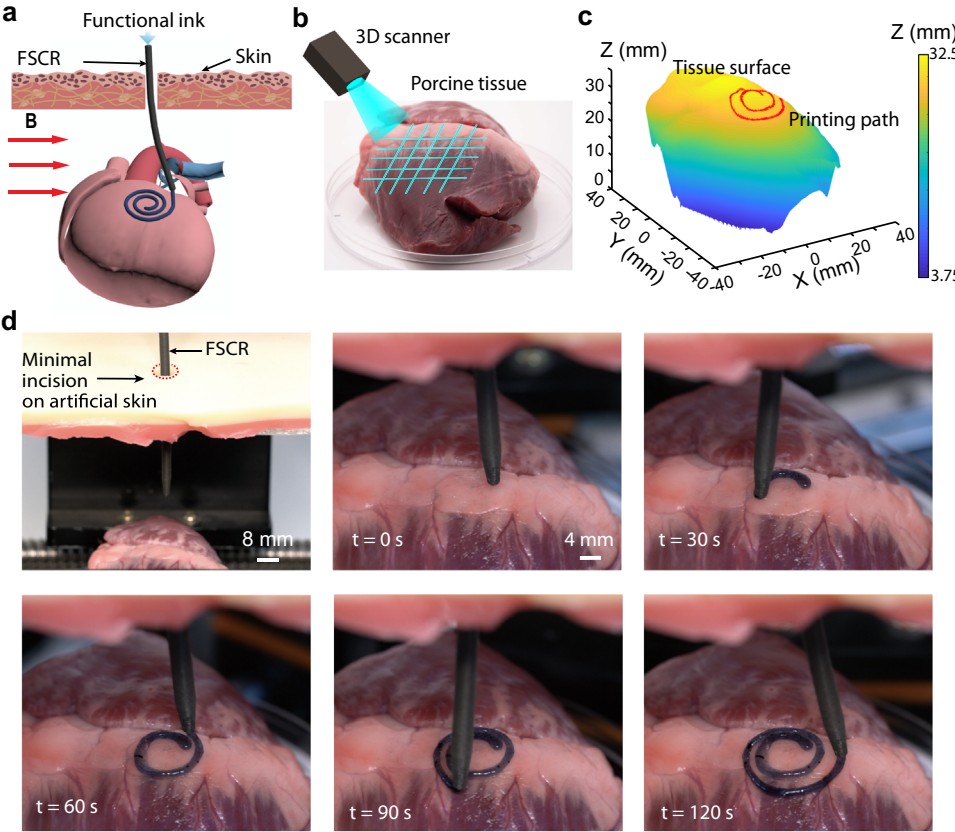

**Fig. 5 In vitro minimally invasive bioprinting of conducting hydrogel on the porcine tissue surface. a** Schematic illustration of the printing process on porcine tissue. **b** 3D scan and reconstruction of the curved surface of porcine tissue. **c** Design of the 3D spiral printing path on the reconstructed surface. The color represents the height magnitude. **d** Photographs of the minimally invasive bioprinting process of a conducting hydrogel on porcine tissue at various times.

the curvilinear and wet tissue surfaces. The major limitations of the potential printable materials originate from how to maintain the as-printed pattern in situ. First, similar to existing extrusion-based biomaterials, the adhesion between the printed material and target surface is critical to shaping the desired pattern. Such a consideration should be taken into account when the target surface is vertical and/or wet. In this regard, enhancing the adhesion between the printed materials and wet bio-tissue surfaces is essential for the quality of bioprinting. Second, injectable inks solidify either through liquid evaporation, gelation, or a temperature-induced or solvent-induced phase change, while minimally invasive bioprinting may not favor such a condition for solidification due to the confined anatomy environment. In particular, when printing complex 3D architectures, the as-printed structure may collapse before it cures. Therefore, reducing the solidification time of the injectable ink and/or employing biodegradable supporting mold to assist solidification is also of great significance. To this end, we envisage that our FSCR and injectable bio-inks with a high adhesive strength to the curvilinear and wet bio-tissues and fast-to-solidify properties will together pave the way for the future applications of minimally invasive bioprinting in a remote, automated, and safer manner.

## Methods

**Fabrication of ferromagnetic soft catheter robot**. The composite ink was made by mixing the hard magnetic NdFeB particle at 15 vol. % with an average diameter of 5 μm into a PDMS matrix (base-to-curing agent at a 10:1 weight ratio, Sylgard 184 silicone, Dow Corning). To ensure homogenous particle dispersion, we stirred the mixture using a planetary mixer (rotation 200 rpm and revolution 2000 rpm, AR-100, Thinky) for 3 min. Ease Release (Ease Release 200, Mann Release Technologies, Inc.) was evenly sprayed on the core wire and mold surface to prevent the

bonding with the elastomer matrix. After that, the mixture slurry was injected into a 3D mold, and a polylactide (PLA) fiber mesh was carefully inserted into the center of the mold together with a 1.0 mm diameter supporting core wire as the inner template. Next, the mold was placed in a vacuum degassing chamber for 1 h to remove the air bubbles and then cured in an oven at 37 °C for 48 h. The cured soft catheter robot was magnetized by a 3850 mT impulse magnetic field generated by a digital pulse magnetizer (Beijing Eusci Technology Ltd).

PLA filaments (average diameter of 150 μm) were employed. Sixteen strands of PLA filaments were knitted into hollow tubes with 1 mm inner diameter by a high-speed automatic knitting machine (Xuzhou Hongtai Knitting Machine Technology Co., Ltd.) at a rate of 20 mm/min.

The basic fabrication process of FSCR with pure PDMS was the same as that was described above. The PDMS-184 base-to-curing agent weight ratio was 8:1. It was cured in an oven at 37 °C for 24 h and then put in an oven at 80 °C for 24 h after demolding.

**Cytotoxicity tests**. Cell survival rate was tested on HCV-29 cell line (Human bladder epithelial cells, American Type Culture Collection). HCV-29 cells were cultured in Roswell Park Memorial Institute (RPMI)-1640 medium (Boster Biological Technology Co. Ltd.) supplemented with 10% fetal bovine serum (Gibco) and penicillin/streptomycin (Boster Biotechnology Co. Ltd.). To investigate the cytotoxicity of NdFeB + PDMS composites, $3 \times 10^3$ cells per well were inoculated in 96-well plates and cultured for 24 h at 37 °C and 5% $CO_2$. The covering material was co-cultured with HCV-29 cells for 24 h without changing the medium. Meanwhile, untreated cells and 70% ethanol-treated cells were employed as positive and negative controls respectively. And the blank well was RPMI-1640 plus CCK-8 reagent. After removing the co-cultured material and replacing the medium, the cell survival rate was evaluated based on the CCK-8 assay according to the manufacturer's protocol (Cell Counting Kit-8, Boster Biological Technology Co. Ltd.). Briefly, 10 μl CCK-8 reagent was added to each well and incubated for 1 h at 37 °C and 5% $CO_2$. The absorbance of each well was measured at 450 nm by a microplate reader (Multiskan FC, Thermo Scientific). Cell survival was calculated by the following formula:

$$\text{Cell survival rate}(\%) = \frac{A_{\text{test}} - A_{\text{blank}}}{A_{\text{control}} - A_{\text{blank}}} \times 100\% \qquad (4)$$

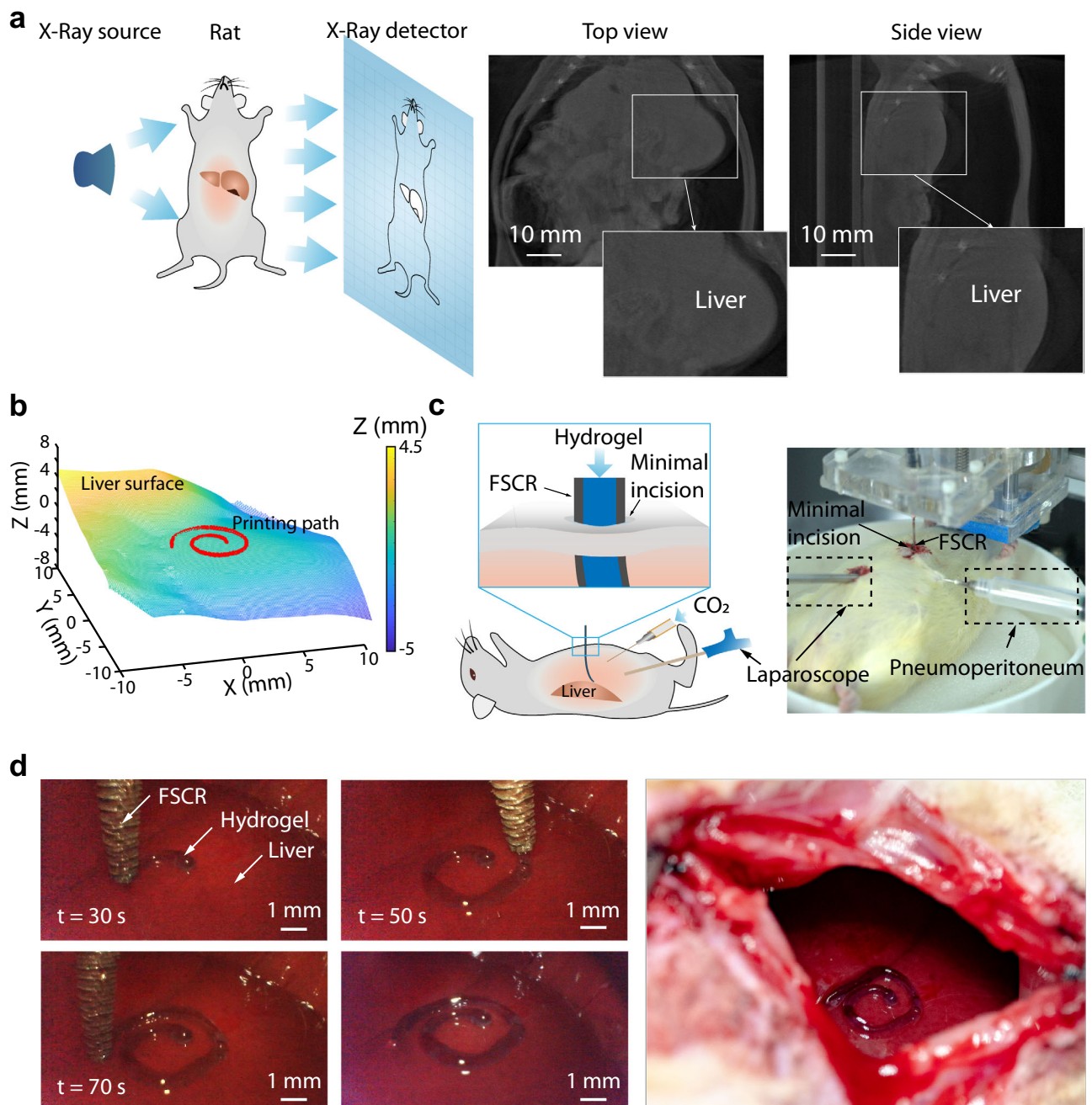

**Fig. 6 In vivo minimally invasive printing of conductive hydrogel on rat liver. a** Left: schematic of CT imaging of a living rat; Right: CT images of a rat with artificial pneumoperitoneum; Insets show profile of liver. **b** Reconstruction of the natural surface of rat liver and desired 3D spiral printing path. **c** Schematic illustration and image of the experimental setup for the in vivo experiment. The color represents the height magnitude. **d** Left: Photographs of the process of minimally invasive bioprinting with a conductive hydrogel on the liver surface at different times; Right and image of the printed pattern on the surface of the rat liver.

where $A_{test}$, $A_{blank}$, $A_{control}$ is tested samples (NdFeB + PDMS composite and 70% ethanol-treated cells), blank controls (blank well), and positive controls (untreated cells), respectively.

**Magnetic characterization**. The magnetic moment densities of ferromagnetic particles were measured with a comprehensive physical property measurement system (Quantum Design) using the vibrating sample magnetometer option. A 7.5 mg of NdFeB powder sample was used. The temperature in the cavity was set to 310 K, the normal temperature of the human body. The maximum symmetrical magnetic field strength was set in 5000 Oe steps from 10000 Oe to 50000 Oe. The hysteresis loop was continuously measured of the same NdFeB powder sample where the magnetic field change rate was set to 200 Oe/sec with a sampling frequency of 1 Hz.

The magnetic field was measured by a precision gauss meter (Multi-Dimension Magnetic Field Scanning and Imaging Test System F30, Beijing Cuihaijiacheng

Magnetic Technology Co., Ltd), which was driven by a moving stage to map the spatial distribution of B in three dimensions.

**Ink preparation**
*Biocompatible viscoelastic ink*. Ecoflex (Ecoflex-0030, PART-A, Smooth On, Inc.) and PDMS-1700 (PDMS SE-1700, Dow, Inc) were mixed to form the printing material. Ecoflex-A, SE-1700 base and SE-1700 curing agent were added at a 10:10:1 weight ratio. To ensure thorough mixing of the particle dispersion, the dispersion was stirred constantly for 3 min using a glass rod. Fat-soluble dye Sudan Red III was added in an amount to aid visualization and the mixture was centrifuged at about $7155 \times g$ for 2 min to remove air bubbles. To avoid changes in the rheological properties of the material, printing should be done promptly after preparation.

*Conductive silver ink*. The polymer solution was prepared by dissolving 5% Alginate powder (Sigma-Aldrich) in deionized water followed by centrifuging for 2 min at a rate of about $7155 \times g$ to remove air bubbles. The conductive ink was prepared by adding silver flakes (with an average diameter of 10 μm, Sigma-Aldrich) and ethanol into the Alginate solution in the weight ratio of 4:6:1. The square resistivity of the ink film was measured with four-point probing equipment (ST2558B-F01, Suzhou Jingge Electronic co., Ltd.). Through wireless power transmission, an alternating magnetic field was generated by an electromagnetic coil with a power of 1600 W. An LED connected to the conductive silver structures was used to demonstrate the inductive currents generated by the alternating magnetic field.

*Hydrogel ink*. Conductive hydrogel ink was prepared by using a physical mixing process in an aqueous solution as previously reported[56,57]. Briefly, 0.1 g Hyaluronic acid, 3 g Pluronic F127 (Energy Chemical), 1.5 g PEDOT: PSS (Clevios™ PH1000, Heraeus Electronic Materials), and 1 g Polycarbophil (Lubrizol) were dispersed in distilled water (gross weight 10 g), and stirred for 24 h in the ice-water bath to minimize foaming. The conductive hydrogel ink was used at room temperature. The two probe testing method was used to test the conductivity of the conducting hydrogel as previously reported[58,59]. Here, the gap *L'* between the two glass carbon electrodes, the inner diameter *D'* of cylindrical mold, and the diameter of electrode *d'* were 3 mm, 6 mm, and 3 mm, respectively. The impedance of the conducting hydrogels was recorded at 5 mV over a range of frequencies from $10^{-2}$ to $10^5$ Hz. Drug (ASA)-loaded hydrogel ink was prepared by using a physical mixing process according to refs. [42,56]. Briefly, 0.1 g Hyaluronic acid, 3 g Pluronic F127, 0.7 g ASA (Energy Chemical) were dispersed in distilled water (gross weight of 10 g) and stirred for 24 h in the ice-water bath to minimize foaming. The ASA-loaded hydrogel ink was used at room temperature. According to references[56,57,60], all constituent materials of the hydrogel were biocompatible."

## Mechanical testing

*Modulus test*. Samples were molded with ferromagnetic composite ink and then cut into dumbbell test specimens using a standard part cutter. The mechanical testing was subjected to standard test methods (ASTM D412) on a mechanical testing machine at a displacement rate of 4 mm/min (width: 4 mm; gauge length: 10 mm).

*Mechanical testing of FSCR*. The lateral load of the reinforced catheter robots and non-reinforced catheter robots (Outer diameter 4 mm, inner diameter 1 mm, length 10 mm) were tested in the same mechanical testing machine as above at a rate of 4 mm/min. Longitudinal tensile strength measurements of the reinforced catheter robots and non-reinforced catheter robots (Outer diameter 4 mm, inner 1 mm, test distance 60 mm) were tested using the same condition.

## Rheology measurements

The rheological properties of the printing inks were measured via a hybrid rheometer (DISCOVERY HR-1, TA Instruments) with a 40-mm diameter rotor. Complex moduli including storage modulus *G'* and loss modulus *G"* of inks were measured using small amplitude oscillatory shear tests over an angular frequency range of 0.1-100 rad/s with an oscillatory strain of 0.1 in the linear viscoelastic region. Apparent shear viscosity was obtained by steady-state flow tests with a logarithmic sweep of shear rate over the range of 0.1-100/s (Supplementary Fig. 10). All rheological properties in these experiments were measured at 25 °C with 120 s soak time prior to heating.

## In vitro drug release study

The dyed ASA-loaded hydrogel was printed onto a piece of porcine tissue which was immersed in PBS solution (pH 7.4)[56]. The drug release testing was carried out by detecting the salicylic acid of the sampling solution using a UV–vis spectrophotometer (UV-3600 Plus, Shimadzu, Japan). The ASA solution undergoes hydrolysis and produces salicylic acid with intrinsic absorbance (peak height) at 297 nm (Supplementary Fig. 14d).

## Printing process

*Printing procedure*. The prepared inks were loaded into the ink chamber. The chamber was then affixed to the designed printing platform that was connected to the fixed end of the soft catheter robot. The CAD pattern was converted into a printing path in modified G-code to adapt our platform to our control algorithm, in which the established functions $T_{tip} = 0.63 T_{mag}$, $U_{tip} = 0.0028 T_{mag}^2 - 0.007 T_{mag}$ were included. The designed pattern was only related to $T_{mag}$. The external superimposed magnetic field was applied to the ferromagnetic soft catheter robot to reorient the soft robot tip during printing. See Supplementary Video for the non-contact printing process.

*Printer configuration*. The hardware printer was assembled by ourselves. The extrusion of printing material was controlled by a digital pneumatic system (Nordson EFD) that was connected to the motherboard Raspberry Pi 2B via the RS232 protocol. All control programs were home written corresponding to the general G-code.

## Construct a printing path on a curved surface

Cloud of surface points of the porcine tissue (the fresh porcine tissue purchased from the local slaughterhouse) and the rat liver were acquired by a laser line scanner (the scanning service was kindly provided by SCANTECH™) and X-ray microtomography (Trans-PET Discoverist 180), respectively. The random data was used to fit the curved surface and reconstruct the surface. For better printing, the X-ray microtomography model has been smoothed, and all the smoothed data was converted into the coordinate system-based data on our platform using commercial software (XPrograma 4.3, also provided by SCANTECH™). Then the fitted curve surface was sampled with raster at equal intervals of 0.1 mm. The sampled data was made into a datasheet for inquiring the height of the z-axis during printing.

## In vivo animal experiments

In all animal experiments, Sprague Dawley rats, 8–10 weeks of age (Vital River Laboratories) were anesthetized with an intraperitoneal injection (2% pentobarbital, 40 ml/kg). The rat abdomen was insufflated with carbon dioxide using a needle to detach the peritoneum from the abdominal organ before performing X-ray microtomography. For the surgery, a digital laparoscope (diameter 0.5 cm) was inserted into the abdominal cavity via a minimal cut (diameter ~0.8 cm) in the left abdomen, and a purse-string suture was tightened around the laparoscope to avoid leaks in the next pneumoperitoneum procedure. A small incision (diameter ~0.3 cm) above the target printing location was made to insert the ferromagnetic soft catheter robot. After that, a needle connected with an adjustable carbon dioxide pump was punctured into the abdominal cavity to create a stable operating space. Then we performed the printing process on the surface of the rat liver. All the animal experiments were approved by the Animal Care and Use Committee of Tongji Hospital, Tongji Medical College, Huazhong University of Science and Technology, Hubei, China. The approval number was IACUC number: 2389.

## Analysis and simulation

Finite element analysis was conducted by a commercial package Abaqus/Standard 2017. To account for the interaction between magnetic composite with embedded hard-magnetic particle and the external non-uniform magnetic field, we developed a user element (UEL) subroutine based on the continuum framework[37]. The magnetic field around the cubic magnet can be analytically expressed according to reference[38]. The magnetic soft catheter robot was meshed with a sufficiently large number of UEL such that during each iteration of computation, the position-dependent magnetic field **B** and its gradient $\nabla$**B** at each element can be accurately calculated. Thereafter, the magnetic torque $\tau = \mathbf{M} \times \mathbf{B}$ and force $f = (\mathbf{M} \cdot \nabla)\mathbf{B}$ can be implemented by computing the magnetic Cauchy stress $\sigma^{magnetic} = -\mathbf{B} \bigotimes \mathbf{FM}$ where **F** is the deformation gradient and operator $\bigotimes$ represents a dyadic product that takes two vectors to yield a second-order tensor. All simulations are checked with convergence.

**Reporting summary**. Further information on research design is available in the Nature Research Reporting Summary linked to this article.

## Data availability

The data generated in this study have been deposited in the github database under accession code https://github.com/softnano501 or can be requested from the corresponding author.

## Code availability

The code in this study has been deposited in the github database under accession code https://github.com/softnano501 or can be requested from the corresponding author.

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

## Acknowledgements
The authors thank others for any contributions. This work was supported by the National Key Research and Development Program of China (2018YFB1105100).

## Author contributions
J.Z., Youzhou Yang. and C.Z. conceived and designed the research. C.Z., Youzhou Yang, Z.G., Q.W., Y.Z, Yueying Yang, and H.T. performed experiments. L.W., Youzhou Yang, and C.Z. performed the theoretical analysis and numerical calculations. J.W., Youzhou Yang, C.Z., Z.G., X.L., and Q.L. performed the in-vivo experiments. C.Z., Youzhou Yang, L.W, and J.Z. wrote the manuscript with input from all authors. All authors participated in drafting the manuscript, discussion, and interpretation of the data. J.Z. supervised the study.

## Competing interests
The authors declare no competing interests.

**Additional information**

