## [Peer Review File · Nature Communications]

REVIEWER COMMENTS

Reviewer #1 (Remarks to the Author):

The manuscript by Zhou, Yang, et al. introduces an innovative approach that combines extrusion-based printing and magnetic control of catheters to deposit material in the body as a strategy of in vivo printing. The emerging field of in vivo printing offers unique approaches to fabricate complex materials at defined sites in the body and has been limited to date by methods that can use a trigger that penetrates the body limiting the access points within the body. This work seeks to extend this paradigm with a clever extension of technologies used for robotic control of a magnetic catheter. The work is innovative and likely to have future impact and is of interest to the Nature Communications readership; however, prior to publication the following issues should be addressed to improve and clarify the manuscript.

1. In the current demonstration, the printing is limited to low resolution (high diameter) filaments that are extruded at relatively high pressure (~240 kPa). While the authors touch on some aspects of this, additional detail should be included to define the limitations to the printing resolution. Is this related to the fact that the material needs to be extruded through the whole length of the catheter? Is the fabrication of the catheter itself inherently limited to low gauge (high diameter) openings? How is the viscosity of the printed solution related to the resolution that can be achieved?
2. The system is described as a minimally invasive strategy to print within the body moving beyond the state of the art (Urciuolo et al. Nat Biomed End 2020) that relies on light. However, the in vivo demonstration shown here requires insufflation of the abdomen to access the liver tissue and provide freedom of movement of the catheter. This is possible in the abdomen but likely not in many other regions of the body. The authors should comment on the full feasibility of this approach. If insufflation is required, could a standard catheter with less sophisticated control be used to achieve the same printing features? How could this be extended to other regions of the body?
3. The current demonstrations are not convincing for why one would need to apply an in vivo printed material. The authors speculate on some potential applications in the discussion but it would be useful if the in vivo application shown here could relate to something beyond the simple deposition of a spiral of material on an intact organ. Can a drug releasing material be used to locally target the tissue?
4. There is little discussion on the potential materials that can be printed with this technology. What are the design constraints and limitations? Can these constraints be mapped on to existing biomaterials that would be of interest for extrusion-based printing? Is this feasible for some of the commonly used materials for ex vivo 3D bioprinting?
5. The figures are well drawn and help to convey the idea very clearly. However, Figure 1a shows many potential options for the use of this system and none of them are reduced to practice in this work.
6. The organization of the manuscript is helpful and clear, however, at times the writing could be improved. For example, the abstract is misleading, "The FSCR is designed to disperse ferromagnetic particles in a fiber-reinforced polymer matrix." It would be better to state that, "The FSCR was designed by dispersing ferromagnetic particles in a fiber-reinforced polymer matrix." The FSCR is later used to dispense the printing ink.

Reviewer #2 (Remarks to the Author):

Authors report ferromagnetic soft catheter robots for minimally invasive bioprinting. Presents results are very interesting and timely new in soft robotics. Therefore, I strongly recommend a possible publication of this article to Nature Communications, if authors properly address the following issues.

1. Please revise introduction section by making a highlight on the novelty.
2. FSCR is suitable for this study. But, please explain if FSCR can be useful for 3D axis operation.
3. Please give a comment on biocompatibility and toxicity of ferromagnetic and hydrogel.
4. Please additionally explain how to implement an integrated vision system (CT) in real time.

Reviewer #3 (Remarks to the Author):

This manuscript demonstrates an in vivo bioprinting system assisted by a well-designed ferromagnetic soft catheter. The soft catheter is used as the printing nozzle in the direct writing way and enables minimally invasive printing on the non-planar surface of organs, controlled by external magnetic field. The authors describe the design of the PLA reinforced printing nozzle and establish correction algorithm of the displacement of the printing nozzle under stimulation of external magnetic field. They further demonstrate this bioprinting concept in vitro and in vivo, achieving the printing on the liver of a rat. In my perspective, this manuscript is of high novelty and high quality. The experimental data can support their conclusions. Thus I recommend the publication of the manuscript on Nature Communications after a minor revision. The following questions and comments are for the consideration of the authors:

- (1) What is the origin of the significant pressure loss when the printing nozzle becomes soft? The lateral expansion is understandable due to the circumferential deformation of the soft catheter. Is the pressure loss related to the circumferential deformation?
- (2) What is the smallest outer and inner diameter of the soft catheter can the authors achieve? Since the outer diameter determines the size of the incision on the skin, it is a key parameter for the concept of minimally invasive bio-printing in vivo. And the inner diameter determines the printing resolution. The authors are suggested to provide this information at appropriate place of the main text. Even if the resolution is not competitive to traditional directing ink writing printing, it is OK. It is good for the readers to know.
- (3) Why do the authors design 4 magnets to control the printing path?
- (4) Most the prints are 2D structures in the manuscript, what is the difficulty of printing 3D structures? The authors can add some discussions on this aspect.

Responses to Comments on “NCOMMS-20-50309-T”

We sincerely thank three reviewers for thoroughly reading our manuscript. And your constructive comments and suggestions are fully addressed in this letter point-by-point. For the convenience of the reviewers, we marked the corresponding modifications in our revised manuscript in blue font. We believe that the questions raised by the reviewers have highlighted areas in need of further attention and the modifications have contributed to a significant improvement of our manuscript, making it more suitable for publication in *Nature Communications*.

Reviewer # 1

General Comment. The manuscript by Zhou, Yang, et al. introduces an innovative approach that combines extrusion-based printing and magnetic control of catheters to deposit material in the body as a strategy of in vivo printing. The emerging field of in vivo printing offers unique approaches to fabricate complex materials at defined sites in the body and has been limited to date by methods that can use a trigger that penetrates the body limiting the access points within the body. This work seeks to extend this paradigm with a clever extension of technologies used for robotic control of a magnetic catheter. The work is innovative and likely to have future impact and is of interest to the Nature Communications readership; however, prior to publication the following issues should be addressed to improve and clarify the manuscript.

Responses. We greatly appreciate your recognition of the significance and impact of our work.

Comment 1

1. In the current demonstration, the printing is limited to low resolution (high diameter) filaments that are extruded at relatively high pressure (~240 kPa). While the authors touch on some aspects of this, additional detail should be included to define the limitations to the printing resolution. Is this related to the fact that the material needs to be extruded through the whole length of the catheter? Is the fabrication of the catheter itself inherently limited to low gauge (high diameter) openings? How is the viscosity of the printed solution related to the resolution that can be achieved?

Response 1

We thank the reviewer for the constructive comment. To evaluate the influences on the resolution of the printed fiber, we carried out systematic experiments and results are presented in the revised supplementary Figure 11. The following sentences are also added to the revised manuscript.

On Page 8, “Due to the viscoelastic nature, the extruded ink usually has a die-swelling phenomenon (Supplementary Fig. 11a), resulting in a printed fiber with a diameter ad , where d is the inner diameter of the printing nozzle and a is the swelling ratio (40, 41). The resolution of the printed fiber (i.e., ad) of our FSCR mainly depends on four parameters: the moving velocity of the nozzle, the input pressure, the inner diameter of the nozzle d , and the viscosity of the ink. As shown in Supplementary Fig. 11b, a faster moving velocity usually stretches the printed fiber, leading to a smaller ad ; while increasing the input pressure, and the size of the inner diameter and the viscosity will increase ad . Note that to ensure a continuous printed fiber without accumulation or discontinuity, the velocity of the nozzle should be well controlled (40, 41). Based on the

injection molding method, the smallest inner diameter d was achieved as 0.6 mm, which yields resolution αd of 0.53 mm at the moving velocity of 3.3 mm/s, pressure of 240 kPa and viscosity of 339 Pa·s (Supplementary Fig. 12 b and c)”

Supplementary Figure 12. The resolution of printed fiber. (a) Schematic illustration of die-swelling phenomenon after ink extrusion. The FSCR nozzle moves at a velocity of V , and the inks are extruded out of the nozzle at a pressure of P , with inner diameter d . The printing resolution is characterized by the diameter of the extruded fiber, denoted as of αd . (b) Increasing velocity V will reduce αd , while increasing the applied pressure P , (c) the size of inner diameter d of FSCR, the diameter of printed fibers will increase αd .

Supplementary Figure 10. Rheological properties of inks. (a) Logarithmic plot of the storage modulus G' and loss modulus G'' as a function of angular frequency ω . (b) The plot of the apparent viscosity η as a function of shear rate γ for viscoelastic inks. The zero-shear viscosity of printed inks are determined by a power law (or Ostwald Law) fit and yield about 339248 mP·s, 581935 mP·s, 71,170 mP·s and 66980 mP·s for the mixture of PDMS-Ecoflex inks and the silver ink. (c) The plot of shear stress τ as a function of shear rate γ , where the curve of the mixture of PDMS-Ecoflex was fitted by Herschel-Bulkley model, while the curve of silver ink was fitted by a power law.

Comment 2.

The system is described as a minimally invasive strategy to print within the body moving beyond the state of the art (Urciuolo et al. Nat Biomed End 2020) that relies on light. However, the in vivo demonstration shown here requires insufflation of the abdomen to access the liver tissue and provide freedom of movement of the catheter. This is possible in the abdomen but likely not in many other regions of the body. The authors should comment on the full feasibility of this approach. If insufflation is required, could a standard catheter with less sophisticated control be used to achieve the same printing features? How could this be extended to other regions of the body?

Response 2.

Thank you for your questions. First, existing standard catheters (including some magnetically-controllable catheters from companies such as Biosense Webster, USA) mainly require manual operation and are not designed for 3D bioprinting. But it is hard for the existing standard catheter to achieve the same printing features that require complicated and numerical control. Our FSCR, to the best of our knowledge, is the first to demonstrate the capability of direct printing deep inside the body via a remotely controllable and automated manner. Second, compared with the complex control system of the commercial magnetic catheters, our control strategy by employing 4 rigid magnets is relatively simple. To address this point, we provide a comparison chart in Supplementary Table 2 to demonstrate the advantages of our FSCR over existing catheters.

Supplementary Table 2. Comparison between our FSCR and representative commercially available catheters

Design/Model	Image	Functionality	Control mechanism	Control apparatus	Reference
Climber (Terumo, Europe)		Guide	Manual pushing/ twisting	Hand	Terumo Europe product catalog
Navistar (Biosense Webster, USA)		Navigation	Magnetic/manual	CARTO® 3 System (Complex operating system)	Biosense Webster product catalog
ACUSON (AcuNav, USA)		Ultrasound catheter	Magnetic/manual	V-Sono™ ICE catheter manipulator and Genesis RMN System (Complex operating system)	ACUSON product catalog
FSCR		Bioprinting	Magnetic/automated	Four magnets (Simple)	Our work

On Page 3, we added “Compared with existing commercial apparatus for magnetically controllable catheters, our control system by employing four permanent magnets is relatively simple (Supplementary Table 2).”

We also clarify that insufflation is generally required in clinical laparoscopic surgeries [Giorgio Bogani *et al.* (2013) DOI:10.1016/j.jmig.2013.12.091] and minimally invasive bioprinting on the outer surface of most abdominal organs with the help of insufflation is already a big step forward to date. However, since our FSCR

is still in its infancy, automated printing on the other parts of the body may not be applicable at this stage. And we are currently working in this direction to empower our FSCR with enhanced printing capability in different environments. To alleviate the concern of the reviewer (also see responses to your **Comment 5**), we have revised Figure 1 and deleted the overstated potential applications.

Comment 3.

The current demonstrations are not convincing for why one would need to apply an *in vivo* printed material. The authors speculate on some potential applications in the discussion but it would be useful if the *in vivo* application shown here could relate to something beyond the simple deposition of a spiral of material on an intact organ. Can a drug releasing material be used to locally target the tissue?

Response 3.

Thank you for your suggestion. Actually, extensive works have demonstrated that injectable bio-inks such as pH-responsive gels [Liang *et al.* (2019) DOI:10.1016/j.jcis.2018.10.056], *in-situ* forming hydrogel [Dimatteo *et al.* (2018) DOI: 10.1016/j.addr.2018.03.007] and silicone rubber [Rasch *et al.* (2020) DOI: 10.1021/acsbiomaterials.0c00094] have applications in therapeutic drug delivery. Our demonstrations in **Figures 4-6** focus on the *in vivo* printing capability of our FSCR, not limited to a specific injectable bio-ink. To address the concern of the reviewer, we added an experiment in which a hydrogel containing a drug component (acetylsalicylic acid (ASA)) was printed to a target lesion, and validated the drug-releasing results by UV-vis spectrophotometer. Results are presented in the revised supplementary Figure 13 and the following descriptions are added to the revised manuscript.

On Page 8 and 9, “To demonstrate the capability of drug delivery to the target lesion, we carried out an *In Vitro* experiment in which a hydrogel containing acetylsalicylic acid (ASA) was printed to a porcine tissue and validated the released drug by UV-vis spectrophotometer (see Materials and Methods for details, Supplementary Fig. 13c and d) (42)”

On Page 14, Ink Preparation

“Drug (ASA)-loaded hydrogel ink was prepared by using a physical mixing process according to (42, 56). Briefly, 0.1g Hyaluronic acid, 3 g Pluronic F127, 0.7 g ASA (Energy Chemical) were dispersed in distilled water (gross weight of 10 g), and stirred for 24 h in the ice-water bath to minimize foaming. The ASA-loaded hydrogel ink was used at room temperature.”

On Page 15,

“***In Vitro* Drug Release Study.** The dyed ASA-loaded hydrogel was printed onto a piece of porcine tissue which was immersed in PBS solution (pH 7.4) (56). The drug release testing was carried out by detecting the salicylic acid of the sampling solution using a UV-vis spectrophotometer (UV-3600 Plus, Shimadzu, Japan). The ASA solution undergoes hydrolysis and produces salicylic with intrinsic absorbance (peak height) at 297 nm (Supplementary Fig. 13 c and d).”

Supplementary Figure 14. Material delivery based on digital magnetic actuation. (a) Experimental demonstration of suction (4% sodium alginate solution dyed blue) and extrusion (4% sodium alginate solution dyed green) in a pre-defined pathway having different geometric sinks along the path. (b) The FSCR is able to lift up the solid blocks with different geometric shapes (weight from 0.5g to 5g). The location of all the objects is random. (c) Printing dyed acetylsalicylic acid (ASA)-loaded hydrogel in a piece of porcine tissue which is immersed in PBS solution (pH 7.4). (d) Absorbance spectra of salicylic acid (SA). After sampling the mixing solution, the released drug is determined using a UV-vis spectrophotometer. ASA solutions undergo hydrolysis forming SA which intrinsic absorbance (peak height) is 297 nm.

Comment 4.

There is little discussion on the potential materials that can be printed with this technology. What are the design constraints and limitations? Can these constraints be mapped on to existing biomaterials that would be of interest for extrusion-based printing? Is this feasible for some of the commonly used materials for ex vivo 3D bioprinting?

Response 4.

Thank you for your suggestion. In this work, we present a printing technology that is capable of direct ink writing via a remotely controllable and automated manner. This technology features the magnetically controllable soft nozzle, while the ink extrusion is governed by its rheological properties such as viscosity, shear-thinning, and yield stress flow. Therefore, our printing technology can be readily integrated with numerous injectable inks such as hydrogels, elastomers, and conductive pastes.

To address the comments of the reviewer, we added the following discussions in the revised manuscript on Page 10.

“Moreover, progress on functional materials (e.g., biomaterials for gastric ulcer healing (54) and health monitoring (55)) with bioprinting-compatible properties (e.g., rheology and adhesion) will enable the FSCR to print more complex 3D patterns/architectures to the curvilinear and wet tissue surfaces. The major limitations of the potential printable materials originate from how to maintain the as-printed pattern *in situ*. First, similar to existing extrusion-based biomaterials, the adhesion between the printed material and target surface is critical to shaping the desired pattern. Such a consideration should be taken into account when the target surface is vertical and/or wet. In this regard, enhancing the adhesion between the printed materials and wet bio-tissue surfaces is essential for the quality of bioprinting. Second, injectable inks solidify either through liquid evaporation, gelation, or a temperature- or solvent-induced phase change, while minimally invasive bioprinting may not favor such a condition for solidification due to the confined anatomy environment. In particular, when printing complex 3D architectures, the as-printed structure may collapse before it cures. Therefore, reducing the solidification time of the injectable ink and/or employing biodegradable supporting mold to assist solidification is also of great significance. To this end, we envisage that our FSCR and injectable bio-inks with a high adhesive strength to the curvilinear and wet bio-tissues and fast-to-solidify properties will together pave the way for the future applications of minimally invasive bioprinting in a remote, automated, and safer manner.”

Comment 5.

The figures are well drawn and help to convey the idea very clearly. However, Figure 1a shows many potential options for the use of this system and none of them are reduced to practice in this work.

Response 5.

We thank the reviewer for pointing this out. To address this, we have revised Figure 1 in which we have deleted the other potential applications while focusing on the comparison between minimally invasive bioprinting of our FSCR and conventional 3D printing using a rigid nozzle.

Fig. 1. Ferromagnetic soft catheter robot (FSCR) system for minimally invasive bioprinting. (A) Schematic illustration of the minimally invasive printing with functional inks (e.g., conducting polymer and living materials) inside the human body through a small incision; in the zoomed-in panel, **B** represents the magnetic field. (B) Schematic illustration of the traditional printing system using a rigid nozzle with a large incision. (C) Schematic of FSCR that is composed of soft polymer matrix with dispersed hard-magnetic particles and polylactic (PLA) reinforcing mesh. The magnetic polarity is programmed along the FSCR axial direction. (D) The numerical control strategy of FSCR, where operations are instructed by digital data. Maneuvering of the FSCR via rotation and translation of the four permanent magnets.

Comment 6.

The organization of the manuscript is helpful and clear, however, at times the writing could be improved. For example, the abstract is misleading, "The FSCR is designed to disperse ferromagnetic particles in a fiber-reinforced polymer matrix." It would be better to state that, "The FSCR was designed by dispersing ferromagnetic particles in a fiber-reinforced polymer matrix." The FSCR is later used to dispense the printing ink.

Response 6.

Thank you for your suggestion. We have carefully revised our manuscript, checked the grammar, and minimized the typos. Besides, two native English speakers have also gone through our revised manuscript to further improve the writing.

Reviewer #2

General Comment.

Authors report ferromagnetic soft catheter robots for minimally invasive bioprinting. Presents results are very interesting and timely new in soft robotics. Therefore, I strongly recommend a possible publication of this article to Nature Communications, if authors properly address the following issues.

Response.

We deeply appreciate the positive comment on our work.

Comment 1.

Please revise introduction section by making a highlight on the novelty.

Response 1.

Thank you for your suggestion. To clarify the novelty of our robot, we have provided two additional comparison charts (added as **Supplementary Table 1** and **2**). We also revised the introduction and highlighted the novelty. On Page 3,

“In the form of a slender rod-like structure with dispersed hard-magnetic particles, FSCR can reach regions inside the body using remote magnetic actuation, followed by in situ printing of functional inks (Fig. 1A) such as lesion healing creams and electrode gels. Distinct from conventional printing systems with a rigid nozzle (Fig. 1B), our FSCR feathers a magnetoactive soft nozzle that can print over a large workspace through a small incision (Supplementary Table 1). To realize steady extrusion of inks, our FSCR is rationally designed with an embedded reinforcing fiber mesh (Fig.1C), which enables printing of various biocompatible and functional inks including silicones, silver pastes, and conductive hydrogels. A magnetic field is imposed by four numerically controlled motor-driven permanent magnets to achieve both translational and rotational motion of the FSCR (Fig.1D). Compared with existing commercial apparatus for magnetically controllable catheters, our control system by employing four permanent magnets is relatively simple (Supplementary Table 2). Our FSCR can print different patterns using multiple inks on both flat and curved surfaces. For the first time, we demonstrate printing a functional hydrogel on a porcine tissue phantom and the liver of a living rat in a minimally invasive, remotely controllable, and automated manner.”

Comment 2.

FSCR is suitable for this study. But, please explain if FSCR can be useful for 3D axis operation.

Response 2.

Thank you for your comment. As has been shown in **Figure 3A**, we clarify that our FSCR is also capable of 3D axis operation by digitally controlling the motor in z-axis. Translational and rotational movement of the four magnets allow for printing on a 2D plane (i.e., XY plane). Meanwhile, the proximal end of FSCR is attached to a motor in z-axis that can be moved upward/downward such that the distance between the printer nozzle and XY plane can be tuned. Such a 3D axis operation has been demonstrated by printing a 3D multilayered tube and scaffold as shown in **Figure 4C** and **Movie S5**.

Comment 3.

Please give a comment on biocompatibility and toxicity of ferromagnetic and hydrogel.

Response 3.

Thank you for the suggestion. First, PDMS is widely used as the polymer matrix for fabricating bio-medical devices due to its high biocompatibility with no acute cytotoxicity [Khadivi et al. (2019), DOI:10.1002/jbm.a.36696; Hacker et al. (2013) DOI:10.1002/app.40132; Meng et al. (2012), DOI:10.1163/156856208786140355). Ferromagnetic particle (NdFeB) is considered to be slightly cytotoxic [Donohue et al. (1995) DOI: 10.1002/jab.770060110; Hopp et al. (2003), DOI: 10.1023/A:1022931915709). But when it is dispersed in PDMS matrix, the cured PDMS + NdFeB composite is bio-safe and biocompatible [Ya Li et al. (2019) DOI:10.1002/adfm.201904977], complying with ISO 10993-5. To further validate the biocompatibility of our FSCR, we conducted the cell viability test and the following description is added to the revised manuscript.

On Page 4, we add “In addition, the cured NdFeB +PDMS composite has no toxicity based on a cell viability test in which the cell survival rate is 98.6%, suggesting a high biocompatibility of our FSCR according to the standard (70% cell survival rate) of USP (ISO 10993-5) (Supplementary Fig. 3) (32, 34)”

Supplementary Figure 3. Biocompatibility tests of the cured NdFeB +PDMS composite. (a) Images of HCV-29 cells cultured with PDMS+NdFeB composite after 24 h. (b) The cell survival rate is 98.6%, suggesting high biocompatibility of our FSCR according to the standard (70% cell survival rate) of USP (ISO 10993-5).

On Page 12 and 13,

“Cytotoxicity Tests. Cell survival rate was tested on HCV-29 cell line (Human bladder epithelial cells, American Type Culture Collection). HCV-29 cells were cultured in Roswell Park Memorial Institute (RPMI)-1640 medium (Boster Biological Technology Co. Ltd.) supplemented with 10% fetal bovine serum (Gibco) and penicillin/streptomycin (Boster Biotechnology Co. Ltd.). To investigate the cytotoxicity of NdFeB+PDMS composites, 3×10^3 cells per well were inoculated in 96-well plates and cultured for 24 h at 37°C and 5% CO₂. The covering material was co-cultured with HCV-29 cells for 24 hours without changing the medium. Meanwhile, untreated cells and 70% ethanol-treated cells were employed as positive and negative controls respectively. And the blank well was RPMI-1640 plus CCK-8 reagent. After removing the co-cultured material and replacing the medium, the cell survival rate was evaluated based on CCK-8 assay according to

the manufacturer's protocol (Cell Counting Kit-8, Boster Biological Technology Co. Ltd.). Briefly, 10 μ l CCK-8 reagent was added to each well and incubated for 1 h at 37°C and 5% CO₂. The absorbance of each well was measured at 450nm by a microplate reader (Multiskan FC, Thermo Scientific). Cell survival was calculated by the following formula.

$$\text{Cell survival rate(\%)} = \frac{A_{test} - A_{blank}}{A_{control} - A_{blank}} \times 100\%$$

where A_{test} , A_{blank} , $A_{control}$ is tested samples (NdFeB+PDMS composite and 70% ethanol-treated cells), blank controls (blank well), and positive controls (untreated cells), respectively.”

As for the printed materials, numerous biocompatible materials can be used according to different purposes. Here in our work, the hydrogel is based on the physical mixing of hyaluronic acid, Pluronic F-127, fillers Polycarbophil, and PEDOT: PSS, all of which are also known to be biocompatible and non-cytotoxic [Jung et al. (2017), DOI: 10.1016/j.carbpol.2016.08.068; Ferreira et al. (2017), DOI: 10.1016/j.jmbbm.2017.02.016; Střiteský et al. (2018), 10.1002/jbm.a.36314].

On Page 3, we emphasize that “To realize steady extrusion of inks, our FSCR is rationally designed with an embedded reinforcing fiber mesh (Fig.1c), which enables printing of various biocompatible and functional inks including silicones, silver pastes, and conductive hydrogels.”

On Page 14, we added “According to references (56, 57, 60), all constituent materials of the hydrogel were biocompatible.”

Comment 4.

Please additionally explain how to implement an integrated vision system (CT) in real time.

Response 4.

Thanks for your comment. In our current printing system, the CT is used to construct the 3D topology of the tissue. The reconstructed 3D model was later used to define the printing path, i.e., generating the numeric code to print. We do not have a vision system in real-time at this stage. To address your comment, we have added the following discussions in the revised manuscript.

On Page 10, “The current printing system adopts the CT to reconstruct the 3D topology of the tissue that is later used to generate the numerical code to print. In the future, utilizing the intraoperative CT and/or equipping the robot with vision-based sensors (e.g., stereovision and structured-light scanning (51)), the real-time tomography of the tissue can be constructed and augmented reality for real-time bioprinting can be achieved (52). In this regard, a close-loop soft robotic system with feedback based on real-time imaging may further improve the accuracy of printing (53).”

Reviewer # 3

General Comment.

This manuscript demonstrates an in vivo bioprinting system assisted by a well-designed ferromagnetic soft catheter. The soft catheter is used as the printing nozzle in the direct writing way and enables minimally invasive printing on the non-planar surface of organs, controlled by external magnetic field. The authors describe the design of the PLA reinforced printing nozzle and establish correction algorithm of the displacement of the printing nozzle under stimulation of external magnetic field. They further demonstrate this bioprinting concept in vitro and in vivo, achieving the printing on the liver of a rat. In my perspective, this manuscript is of high novelty and high quality. The experimental data can support their conclusions. Thus I recommend the publication of the manuscript on Nature Communications after a minor revision. The following questions and comments are for the consideration of the authors

Response.

Thank you very much for your positive comments.

Comment 1.

What is the origin of the significant pressure loss when the printing nozzle becomes soft? The lateral expansion is understandable due to the circumferential deformation of the soft catheter. Is the pressure loss related to the circumferential deformation?

Response 1.

Thank you for raising this question. We realize that our previous statement on Page 4 and 5 in the main text “In general, due to the nature of the soft nozzle, the pressure loss is significant along the channel” is inaccurate. Instead, the pressure loss is mainly due to the energy dissipation induced by the viscosity and friction between the ink and the wall [Phan et al. (2018) DOI:10.1122/1.5022982]. Then pressure loss requires higher pressure input that causes the circumferential expansion of the soft nozzle, leading to a time delay in printing. Therefore, we have corrected our statement on Page 4 and 5, “In general, due to the ink viscosity and friction between the ink and the nozzle, the input energy dissipates during the process of ink extrusion, leading to a pressure loss along the channel (see Supplementary Materials for details) (35).”

Comment 2.

What is the smallest outer and inner diameter of the soft catheter can the authors achieve? Since the outer diameter determines the size of the incision on the skin, it is a key parameter for the concept of minimally invasive bio-printing in vivo. And the inner diameter determines the printing resolution. The authors are suggested to provide this information at appropriate place of the main text. Even if the resolution is not competitive to traditional directing ink writing printing, it is OK. It is good for the readers to know.

Response 2.

We thank the reviewer for pointing this out. Based on our molding method, the smallest outer and inner diameters in our designs are about 2 mm and 0.6 mm, respectively. Although the printing resolution may not be competitive to traditional direct ink writing printing, such a small size has already complied with the standard incision (5 -10 mm) in minimally invasive surgeries [M.D. Mark Vierra. (1995) DOI:

annurev.med.46.1.147].

On Page 4, “By employing different molds, we can easily fabricate FSCRs of various sizes that can be used in different applications (Fig. 2A) and the smallest outer and inner diameter can be achieved as 2 and 0.6 mm, respectively, complying with the standard of incision size in minimally invasive surgeries (33).”

Comment 3.

Why do the authors design 4 magnets to control the printing path?

Response 3.

Thank you for your question. Two major considerations are taken into account when designing the magnetic field: (1) how to maintain the FSCR straight without any deformation before printing; (2) how to increase the stability of the control. To answer this, we added the following reasons in the revised manuscript.

On Page 6, “The primary reasons for designing such a 4-magnet control system are twofold. First, due to symmetry, the magnetic field has $B_x = B_y = 0$ along the z-axis and in particular $\mathbf{B} = \mathbf{0}$ at the origin. Thus, the FSCR is at equilibrium initially when the entire body is aligned with the z-axis and the tip coincides with the origin, maintaining a vertical configuration before printing. Second, placing two magnets with north poles facing each other in both x- and y-direction allow for more stable control of the FSCR.”

Comment 4.

Most the prints are 2D structures in the manuscript, what is the difficulty of printing 3D structures? The authors can add some discussions on this aspect.

Response 4.

Thank you for your comments. In addition to the 2D structures, we have already demonstrated printing a 3D five-layered tube. To further demonstrate the capability of printing 3D structures, we added a 3D five-layered scaffold in the revised Figure 4 in the main text. The corresponding video is shown in Supplementary Movie 5.

Fig. 4 Invasive direct ink printing by an FSCR. b Printing 3D five-layered tube and scaffold with viscoelastic materials.

We also add following discussions to the revised manuscript.

On Page 10, “Moreover, progress on functional materials (e.g., biomaterials for gastric ulcer healing (54) and health monitoring (55)) with bioprinting-compatible properties (e.g., rheology and adhesion) will enable the FSCR to print more complex 3D patterns/architectures to the curvilinear and wet tissue surfaces. The major limitations of the potential printable materials originate from how to maintain the as-printed pattern *in situ*. First, similar to existing extrusion-based biomaterials, the adhesion between the printed material and target surface is critical to shaping the desired pattern. Such a consideration should be taken into account when the target surface is vertical and/or wet. In this regard, enhancing the adhesion between the printed materials and wet bio-tissue surfaces is essential for the quality of bioprinting. Second, injectable inks solidify either through liquid evaporation, gelation, or a temperature- or solvent-induced phase change, while minimally invasive bioprinting may not favor such a condition for solidification due to the confined anatomy environment. In particular, when printing complex 3D architectures, the as-printed structure may collapse before it cures. Therefore, reducing the solidification time of the injectable ink and/or employing biodegradable supporting mold to assist solidification is also of great significance. To this end, we envisage that our FSCR and injectable bio-inks with a high adhesive strength to the curvilinear and wet bio-tissues and fast-to-solidify properties will together pave the way for the future applications of minimally invasive bioprinting in a remote, automated, and safer manner.”

REVIEWER COMMENTS

Reviewer #1 (Remarks to the Author):

The authors have nicely improved the manuscript in their thoughtful response to all of the reviewers' comments. The manuscript is now suitable for publication in Nature Communications.

Reviewer #2 (Remarks to the Author):

The revised paper is now acceptable.

Reviewer #3 (Remarks to the Author):

The authors have done a good job of revising the manuscript, according to the first round review. They have added more experiments to justify and clarify their statement. All my questions are addressed adequately. Thus I recommend the publication of the paper.